# Combination of Poly(ε-Caprolactone) Biomaterials and Essential Oils to Achieve Anti-Bacterial and Osteo-Proliferative Properties for 3D-Scaffolds in Regenerative Medicine

**DOI:** 10.3390/pharmaceutics14091873

**Published:** 2022-09-05

**Authors:** Sara Comini, Sara Scutera, Rosaria Sparti, Giuliana Banche, Bartolomeo Coppola, Cinzia Margherita Bertea, Gabriele Bianco, Noemi Gatti, Anna Maria Cuffini, Paola Palmero, Valeria Allizond

**Affiliations:** 1Department of Public Health and Pediatrics, University of Torino, Via Santena 9, 10126 Turin, Italy; 2Department of Applied Science and Technology, Politecnico di Torino, Corso Duca degli Abruzzi, 24, 10129 Turin, Italy; 3Department of Life Sciences and Systems Biology, University of Torino, Via Quarello 15/A, 10135 Turin, Italy; 4Microbiology and Virology Unit, Azienda Ospedaliero Universitaria Città della Salute e della Scienza di Torino, Corso Bramante 88/90, 10126 Turin, Italy

**Keywords:** biomaterial associated infection, poly(ε-caprolactone)-based biomaterial, essential oils, *Staphylococcus aureus*, *S. epidermidis*, *Escherichia coli*, anti-adhesive/anti-bacterial properties, Saos-2 cells’ cell viability/proliferation

## Abstract

Biomedical implants, an essential part of the medical treatments, still suffer from bacterial infections that hamper patients’ recovery and lives. Antibiotics are widely used to cure those infections but brought antibiotic resistance. Essential oils (EOs) demonstrate excellent antimicrobial activity and low resistance development risk. However, EO application in medicine is still quite scarce and almost no research work considers its use in combination with bioresorbable biomaterials, such as the poly(ε-caprolactone) (PCL) polymer. This work aimed to combine the antibacterial properties of EOs and their components, particularly eugenol and cinnamon oil, against *Staphylococcus aureus*, *S. epidermidis* and *Escherichia coli*, with those of PCL for medical applications in which good tissue regeneration and antimicrobial effects are required. The PCL porous scaffolds, added with increasing (from 30% to 50%) concentrations of eugenol and cinnamon oil, were characterized by square-shaped macropores. Saos-2 cells’ cell viability/proliferation was hampered by 40 and 50% EO-enriched PCL, whereas no cytotoxic effect was recorded for both 30% EO-added PCL and pure-PCL. The antibacterial tests revealed the presence of a small inhibition halo around the 30% eugenol and cinnamon oil-functionalized PCL scaffolds only for staphylococci, whereas a significant decrease on both adherent and planktonic bacteria was recorded for all the three microorganisms, thus proving that, even if the EOs are only in part released by the EO-added PCL scaffolds, an anti-adhesive feature is anyway achieved. The scaffold will have the ability to support new tissue formation and simultaneously will be able to prevent post-surgical infection. This research shows the great potential in the use of EOs or their single components, at low concentrations, for biomaterial functionalization with enhanced anti-bacterial and biointegration properties.

## 1. Introduction

Antibiotics, since their discovery, have been used initially for human healthcare; thereafter, the consumption has been extended to animal therapeutics, agriculture, livestock, and food security [1,2,3]. Unfortunately, the wide-spread distribution, the excessive use and the misuse of these antimicrobial agents determined the intensification of antibiotic-resistant microorganisms, which, nowadays, constitute one of the relevant challenges to global health, rendering the treatment of microbial infections more problematic and costly than it has ever been [4,5,6,7,8,9,10]. The antimicrobial resistance is due to the development of genetic and biochemical mechanisms that allow microorganisms to survive in antibiotic environments [11]. The increasing microbial resistance to conventional antimicrobial agents has encouraged scientists to search for novel sources of molecules with broad-spectrum activities [9,12,13].

Orthopedic devices are necessary in different parts of the human body to substitute the functions of lost or disabled joints and to allow tissue healing. However, the implant structure tightly interacts with biological fluids, thus offering an appropriate habitat for bacteria or fungi to adhere and proliferate, leading to serious infections and impeding its function, also causing an inflammation environment [14,15,16,17]. Various microorganisms determine orthopedic infections, mainly Gram-positive cocci e.g., *Staphylococcus aureus* and coagulase-negative *Staphylococcus* species, anaerobes, and fungi. Gram-negative bacteria, such as *Escherichia coli,* have also been implicated in orthopedic infections [13,15,18,19]. Furthermore, the ability of bacteria to form biofilm on medical devices makes combating them even more challenging, thus the prevention of biofilm formation is considered preferable to its removal, since the latter is a very difficult and demanding task [2,5,6,20,21,22].

For these reasons, unconventional approaches with respect to antibiotics are necessary to treat infections. Natural plant-derived products have been used for many generations in both folk and traditional medicine [4]. In this context, essential oils (EOs) are mixtures of volatile compounds extracted from aromatic plants and may constitute a promising source for new natural drugs [11,23]. EOs are the largest group of secondary metabolites produced by plants formed from tens to hundreds of molecules belonging to the class of terpenoids and phenylpropane derivatives. The most common constituents in volatile oils are monoterpenes, sesquiterpenes and phenolic compounds with oxygenated or non-oxygenated derivatives [4,5,7,24,25]. There is an increasing amount of attention for EOs in nanotechnology since outstanding biological and medical features characterize them, in particular, bactericidal, virucidal, fungicidal, antiparasitical, insecticidal, analgesic, anti-inflammatory, spasmolytic, anesthetic, and anti-oxidative properties [2,5,6,9,23,25,26]. Even if the exact mechanisms associated with the antibacterial activity of EOs is still incompletely identified, different actions have been recently suggested in literature that are the changes of bacterial membrane permeability by altering both transport systems and energy production [7,27]. Thanks to their multi-component nature, a multi-target antimicrobial action of EOs has been suggested, and since today, there has been no confirmation of EO resistance manifestation [1,2,23,28,29].

Poly(ε-caprolactone) (PCL), a polyester polymer having a semicrystalline structure, is a biocompatible, bioresorbable and low cytotoxic on human cells biomaterial that has been approved by the Food and Drug Administration (FDA) for use in several medical and drug delivery devices [1,6,15,23,30,31,32]. To create a controlled release system of EOs from PCL, the incorporation into the biomaterial is suggested [33].

The main objective of the present research was to design and develop novel modified PCL three-dimensional (3D) scaffolds for potential applications in the biomedical field-with antimicrobial properties by adding eugenol and cinnamon EO, as antimicrobial agents. The evaluation of the influence of these molecules on: (i) PCL morphological characteristics; (ii) cytotoxicity behavior on human sarcoma osteogenic-2 (Saos-2) cells; and (iii) anti-adhesive properties against Gram-positive and Gram-negative bacteria were carried out.

## 2. Materials and Methods

### 2.1. PCL-Based Scaffold Preparation and Characterization

The pellets of poly(ε-caprolactone) (PCL, molecular weight 80,000, density of 1.145 g/cm^3^; Merck KGaA, Darmstadt, Germany) were dissolved in acetone (acetone–PCL weight ratio 80:20) at 40 °C for 24 h. Thereafter, different eugenol and cinnamon oil increasing percentages (30, 40 or 50%) by weight with respect to PCL were added under magnetic stirring for 15 min at room temperature. Specifically, eugenol (eugenol nat M_0020312, purity (GC) >98%) or cinnamon (cinnamon leaf rectified essential oil M_0058821) EO—kindly provided by MANE (www.mane.com, accessed on 11 February 2019)—were used as antimicrobial agents. Then, sodium chloride (NaCl) or sodium nitrate (NaNO_3_) granules (>99.5% purity, Merck KGaA) were used as the template to generate the porosity in the scaffolds. The salts were sieved in the 125–355 µm range before being added to the mixture (NaCl or NaNO_3_–PCL weight ratio 90:10, to produce a microporosity of about 80 vol%). As reported in a previous paper [30], NaCl granules were characterized by regularly shaped grains, most of them cubic, with an average size of about 330 μm. On the other hand, NaNO_3_ granules were characterized by less defined geometry and smaller grains (average size of about 270 μm) with more rounded edges. Next, the obtained mixture was cast into cylindrical plastic molds (20 mm diameter, 10 mm height). Samples were dried into a closed chamber for 3 days, then demolded and immersed in deionized water for 5 days to leach out the salt crystals; the deionized water was changed daily. Control specimens of pure PCL without eugenol and cinnamon oil were also fabricated according to the procedure previously described [30]. Hence, different scaffold types were obtained: pure PCL, PCL + eugenol or cinnamon EO at 30 (PCL eug 30% or PCL cin 30%), 40 (PCL eug 40% or PCL cin 40%) or 50% (PCL eug 50% or PCL cin 50%) by weight with respect to PCL, with NaCl or NaNO_3_ salts as pore formers. 

All the prepared PCL-based biomaterials were characterized for diameter (mm), height (mm) and density (mg/mm^3^), and the results were reported as mean values ± standard error of the mean (SEM). The morphology of the 3D scaffolds was acquired by means of Field Emission Scanning Electron Microscopy (FESEM, Zeiss Supra 40, Jena, Germany).

To quantify the EO release from the eugenol or cinnamon oil-doped PCL-based specimens, during the washing of the salt leaching process, the wash water was first extracted with hexane and then qualitatively and quantitatively analyzed by Gas Chromatography (GC) coupled with Mass Spectrometry (MS). Briefly, n-hexane containing 1 mg mL^−1^ iso-thymol as internal standard, was added to the samples in a 1:1 (*v*/*v)* ratio. The samples were vigorously mixed in a separator funnel and the organic upper phase was then removed and concentrated to 1 mL using N_2_ flow before injection into GC-MS.

The analysis was carried out using an Agilent 6890N gas chromatograph equipped with an HP5-MS column (30 m length, 250 μm diameter, 0.25 μm thickness) and coupled to an Agilent 5973A mass spectrometer (Agilent Technologies, Santa Clara, CA, USA). Helium (1 mL min^−1^) was employed as carrier gas and the chromatographic separation conditions were: 60 °C as initial temperature held for 1 min followed by a ramp of 3 °C per min^−1^ up to 200 °C. This final temperature was held for an additional 3 minutes. Mass spectra were acquired in full scan mode (range of *m*/*z* 50–220). In total, 1 μL of each sample was injected in the GC-MS, in split less mode, and the EO components were identified by comparing the mass fragmentation spectra with the reference NIST 98 software and quantified using the standard iso-thymol.

### 2.2. Degradation Rate of PCL-Based Scaffolds

The degradation rate (DR) degree assay was performed with the immersion of the PCL-based construct, with or without eugenol and cinnamon oil, in a solution of simulated body fluid (SBF) at 37 °C during different incubation times (3, 6, 12 and 18 days). The SBF medium—specifically, HEPES-free Dulbecco’s Modified Eagle Medium with 10% fetal bovine serum—containing an ion concentration almost equal to those of human blood plasma, was prepared in our laboratory with substances purchased from Sigma Aldrich (now Merck KGaA) [11,34,35]. At each time point, the 3D scaffolds were removed from the SBF, kept under a laminar flow until they reached a constant mass and then weighed. The DR values of the pure PCL and of the EO-added PCL-bases scaffolds were determined with the following formula:$$\mathrm{DR}\;\left(\%\right)=\frac{m_{0}-m_{x}}{m_{0}}\times 100$$
where:m_x_—mass of the PCL-based constructs after drying; m_0_—initial mass of the PCL-based constructs.

### 2.3. Cell Viability Assay by Direct-Contact Test

The in vitro direct-contact experiments were performed with the human osteosarcoma cell line Saos-2 (American Type Culture Collection ATCC^®^, HTB-85) that exhibits an osteoblast-like phenotype. As previously detailed [30], after Saos-2 cells growth in a high-glucose Dulbecco’s modified minimum essential medium (DMEM) with phenol red and supplements, 2 × 10^4^ cells were seeded onto the surface of the sterile PCL-based biomaterials, with or without eugenol and cinnamon oil, that were previously sterilized and cut into small cylinders (5 mm in diameter and 5 mm in height) in 96-well plates and cultured in 200 µL cell culture medium for 0, 3, 6, and 12 days. Controls of pure-PCL samples and EO-added constructs were assayed in triplicate. The medium was substituted every 2–3 days. Cell viability was evaluated at the different time points by the 3-(4,5-Dimethylthiazol-2-yl)-2,5-Diphenyltetrazolium Bromide (MTT) assay (Merck KGaA), and the optical density (OD) was measured at 570 nm using a microplate reader (VICTOR3TM, PerkinElmer, City, MA, USA).

At each time point, data were expressed as cell viability percentages (%) according to the following formula: $$\mathrm{cell}\;\mathrm{viability}\;\left(\%\right)=\frac{{\mathrm{OD}}_{\mathrm{PCL}-\mathrm{Saos}-2}-{\mathrm{OD}}_{\mathrm{PCL}}}{{\mathrm{OD}}_{\mathrm{Saos}-2}}\times 100$$
where:OD_PCL-Saos-2_: OD of the PCL-based scaffolds with Saos-2 cells;OD_PCL_: OD of the PCL-based scaffolds without Saos-2 cells;OD_Saos-2_: OD of the Saos-2 cells.

The Saos-2 cell morphology and adhesion to PCL-based scaffolds, at the same time points, was additionally assessed by means of FESEM (Zeiss Supra 40) analysis.

### 2.4. In Vitro Antibacterial Tests

Three different strains, *S. aureus* (ATCC 29213), *S. epidermidis* (ATCC 35984), and *E. coli* (ATCC 25922), as leading bacteria involved in orthopedic infections, were employed for antibacterial tests. The inhibition halo assay in accordance with EUCAST guidelines (manual v 9.0; https://www.eucast.org/ast_of_bacteria/disk_diffusion_methodology, accessed on 13 January 2020) and the bacterial adhesion experiments were performed to evaluate the antimicrobial activities of the EO-enriched PCL-based 3D scaffolds [30,36].

#### 2.4.1. Inhibition Halo Assay

The antimicrobial activity of eugenol and cinnamon oil released from added samples was determined by the inhibition halo test. According to the method previously described in our study [30], a bacterial suspension of ~1–2 × 10^8^ colony forming units (CFUs/mL)—corresponding to 0.5 McFarland density—was spread using a sterile swab on Mueller Hinton Agar (MHA, Becton Dickinson and Company, BD, Franklin Lakes, NJ, USA) plates. Subsequently, the sterilized PCL-based samples, with and without EOs, were placed and carefully pressed on the surface of the agar and incubated for 24 h at 37 °C. Thereafter, the diameter of the area around the samples where the bacterial growth stopped was measured (mm).

#### 2.4.2. Bacterial Adhesion Assay

As described in detail in our previous research [30,36], the evaluation of the bacteria ability to adhere to the surface of the PCL-based scaffolds, with and without EOs, was performed at different incubation times. Briefly, the overnight bacterial cultures of *S. aureus*, *S. epidermidis* and *E. coli* were diluted in Mueller Hinton Broth (MHB; Becton Dickinson and Company) to ~10^4^ CFU/mL. Subsequently, the sterile biomaterials were covered with 7 mL of the bacterial suspension and incubated at 37 °C by shaking for 7 h and 24 h. After the incubation period, a sonication protocol—40 kHz for 30 min at 22 °C in 10 mL of sterile saline solution (Bieffe Medital S.p.A., Grosotto, Italy)—was used to quantify (as CFU/mL) the 3D scaffold-bound bacteria. The number of CFU was determined by serial plate count of each sonication product after a 10 min centrifugation at 4000 rpm and subsequent concentration in 1 mL of sterile saline solution (Bieffe Medital S.p.A.). The number (CFU/mL) of planktonic bacteria was also determined. All the experiments were executed simultaneously for each sample, assayed in triplicate and repeated a minimum of three times. 

### 2.5. Statistical Analysis

The weight loss, as degradation rate percentages, the MTT test results, as cell viability percentages and microbiological data, as log_10_ CFU/mL, were analyzed by descriptive statistics (mean values and standard error of the mean) and tested by an unpaired Student’s *t*-test by the GraphPad Prism 9 software (San Diego, CA, USA). The rate of significance was settled at *p* < 0.05.

## 3. Results

### 3.1. PCL-Based Biomaterial Morphological Characterization

The combinations of the synthetic polymer, PCL, and eugenol and cinnamon oil were prepared as oil-in-acetone emulsions by emulsifying EOs, at increasing concentrations, into a solution containing the dissolved polymer; thereafter, the porous agent was added (NaCl or NaNO_3_) and the samples casted into the molds. As reported in detail in Table 1 and Table 2, all the prepared PCL-based biomaterials added with eugenol and cinnamon oil had a cylindrical geometry and similar dimensions, specifically diameter (mm) and height (mm), independent of the porous agent used ( NaCl or, NaNO_3_).

Pure-PCL samples were characterized by a density of 0.126 ± 0.003 g/cm^3^ (when fabricated with NaCl as the pore former) and of 0.136 ± 0.002 g/cm^3^ (with NaNO_3_). On the ground of the density of neat polycaprolactone, the scaffolds showed a total porosity of 89.0% (NaCl) and 88.1% (NaNO_3_), providing a very good agreement with the nominal porosity (90%), as indicated in Materials and Methods section (see Section 2.1).

On the contrary, the EOs addition—at 40% or at 50%—determined a significant (*p* < 0.001) increase in both weight (mg) and density (mg/mm^3^) of the samples with respect to pure PCL scaffolds, but independently to the type of EO, eugenol or cinnamon, used (Table 1). In particular, the density of pure-PCL samples, pored with NaCl, increased to ~0.300 g/cm^3^ when 40–50% of eugenol or cinnamon were added to the polymer (Table 1). A similar augmented weight and densitiy pattern was also observed for the scaffold prepared with NaNO_3_ as pore-forming salt (Table 2).

The aspect of the PCL based scaffolds is presented in Figure 1: the presence of the EOs did not determine a change in the general morhpology of the constructs.

Figure 2 illustrates some demonstrative FESEM micrographs of the sections of porous pure-PCL constructs, obtained by using both NaCl (Figure 2A) and NaNO_3_ (Figure 2B) pore formers. The morphology and size of the pores in the scaffolds were affected by those of the salts used as the templates. In Figure 2, we can observe that the scaffolds produced with NaCl granules were characterized by well-defined geometrical pores, with an average size of 234 ± 61 μm (as determined on a cut section of the scaffold); on the other hand, the scaffolds produced starting from NaNO_3_ presented pores with a less defined geometry, with an average size of 208 ± 34 μm. In both cases, we can observe that the size of the pores in the scaffolds was smaller than that of the starting templating salts, indicating a certain shrinkage of the scaffolds during drying. The morphology of the 30% EO-functionalized scaffolds is depicted in Figure 2C,D, for 30% eugenol-enriched PCL, and in Figure 2E,F, for 30% cinnamon oil-added PCL constructs. As clearly evident, the geometrical pores due to the salt leaching process, and a high-interconnected porosity can be easily observed, suggesting that the 30% eugenol and cinnamon oil addition did not affect the 3D scaffold features. In Figure 3, a higher magnification FESEM image representative of 30% eugenol or cinnamon oil-enriched PCL scaffolds (A, NaCl and B, NaNO_3_) demonstrates the presence of micropores within the struts and the pore walls (white arrows), thus creating an open and interconnected porosity, as those revealed for pure-PCL scaffolds. Finally, Figure 4 shows the microporosities of the biomaterial surfaces, particularly those of 30% eugenol-added PCL samples by using NaCl (A) or NaNO_3_ (B), thus demonstrating that the highly interconnected pores are present not only in the deep levels of the scaffolds but in their surfaces too. The samples prepared at 40% and 50% EO showed a comparable microstructure to those of lower EO content (FESEM images not shown). 

The quantification, by GC-MS analysis, of the content of eugenol and cinnamon oil on the washing water, during the salt-leaching phase of the sample preparation, revealed that from the first to the fifth wash about 20% of the eugenol or cinnamon EO was released by the constructs, for both salts used to form the pores. These data indicated that the active concentration of EOs within the 3D scaffolds was reduced to <10% for the 30% EO-enriched PCL-based samples, and similar reduced percentages were obtained for the 40% and 50% EO-added ones.

### 3.2. PCL-Based Scaffold Degradation Degree 

The weight variation, expressed as degradation rate percentages, of pure-PCL or PCL-based enriched with eugenol and cinnamon oil 3D scaffolds, during the immersion in SBF for 3, 6, 12 and 18 days of incubation, is reported in Figure 4. Starting from three days in SBF, the constructs underwent a significant (*p <* 0.05) decrease in weight reduction, reporting a mean DR% of 3.76% or 6.47%, for pure-PCL/30% EO-added or 40%/50% EO-enriched PCL samples, respectively. Thereafter, a constant weight loss, within 12 days, was obtained for the sample of pure-PCL and of 30% EO-added maintaining values of about 3.7%. On the contrary, a significant (*p <* 0.001) reduction in weight was recorded for the 40% and 50% added-PCL based scaffolds, with DR% values of ~14.15%. At the end of the assay—after 18 days of immersion into SBF—an increased weight loss was recorded for all the constructs, as detailed in Figure 5.

At each time point, the 3D constructs were analyzed for morphology modification, by using a FESEM approach, and no alteration in the high-interconnected porosity was observed (data not shown).

### 3.3. In Vitro Saos-2 Cell Viability/Proliferation Assay

Since the scaffolds were designed for tissue regeneration purposes, the evaluation of the potential influence of increasing concentrations of eugenol and cinnamon oil added to the PCL-based biomaterials on Saos-2 cell viability and proliferation was performed by using MTT assays and FESEM analysis, within 12 days of incubation.

The effect of eugenol and cinnamon oil from 30% to 50% inside the PCL-based constructs on Saos-2 cell viability, evaluated as cell viability percentages, is shown in Figure 6. In detail, at the beginning of incubation (0 day), a similar pattern of cell viability percentages was obtained for all the PCL-based scaffolds, with no differences between the control and EO-enriched biomaterials. After 3 days of incubation an increase in Saos-2 cell proliferation was obtained for all the PCL-based scaffold, although a significant reduction into cell viability was detected for the 40% and 50% EO-added samples (*p* < 0.05) with respect to both pure-PCL and 30% EO-added constructs. After six days of incubation (Figure 6), a higher cell viability percentage has been observed for both pure-PCL and 30% eugenol (Figure 6A) or 30% cinnamon oil (Figure 6B)-added biomaterials. On the contrary, at the same incubation time, a significant (*p* < 0.001) reduction in Saos-2 cell viability was revealed for the 40% and 50%-added PCL scaffolds, with eugenol or cinnamon oil (Figure 6). A slight reduction into cell viability percentages was recorded, after 12 days of incubation, for pure-PCL and 30% EO-added specimens (~70%), whereas values <10% of cell viability for the 40–50% PCL-enriched constructs were obtained (Figure 6).

The FESEM analysis on the 30% eugenol and 30% cinnamon oil-added PCL-based biomaterials demonstrated the attachment of osteoblast-like cells on the 3D scaffolds, at six days of incubation (Figure 7). It is worth noting that, within 12 days of incubation, Saos-2 cells not only adhered to the 3D construct but also integrated on the surface (Figure 8), thus confirming that the 30% eugenol and cinnamon oil enrichment of the constructs did not interfere negatively with eukaryotic cells attachment and proliferation into the biomaterial. 

### 3.4. Antibacterial Assays

All the microbiological assays were performed by using the 30% eugenol and cinnamon oil concentration into the PCL-based biomaterials that demonstrated themselves to be non-toxic for eukariotic cells, since the addition of 40% and 50% of either eugenol or cinnamon oil determined a relevant reduction into Saos-2 cell viabilitiy and proliferation.

The inhibition halo assay revealed the release of either 30% eugenol or 30% cinnamon oil from the PCL-based EO-added samples, since it was accompanied by a halo of bacterial growth inhibition close to the scaffolds. In fact, as reported in Figure 9 for the two Gram-positive microorganisms, an inhibition halo was detected. A higher inhibition halo with 30% cinnamon oil—for both *S. aureus* (Figure 9B, Ø = 28.63 ± 0.17 mm) and *S. epidermidis* (Figure 9D, Ø = 27.71 ± 0.36 mm)—was observed with respect to 30% eugenol (Figure 9A, Ø = 21.07 ± 0.30, and 8C, Ø = 24.61 ± 0.24 mm, respectively). On the contrary, concerning the Gram-negative bacterium *E. coli*, no inhibition halo was observed on PCL EO-added samples irrespective to the EO added (Figure 9E,F). These data were similar regardless of the salt (NaCl or NaNO_3_) used to produce the pores of the scaffolds.

The adhesion assay results obtained for both Gram-positive (*S. aureus* and *S. epidermidis)* and Gram-negative (*E. coli)* bacteria, as a number of the adherent microorganisms (log_10_ colony forming unit, CFU/mL) on PCL-based scaffolds added with the 30% of the EOs eugenol or cinnamon oil, produced with either NaCl or NaNO_3_ salts, were achieved after 7 and 24 h of incubation. 

Specifically, after 7 h of incubation, the *S. aureus* adhesion on the control materials, pure-PCL, was 1.52 × 10^6^ ± 5.15 × 10^5^ CFU/mL and 1.04 × 10^6^ ± 2.61 × 10^5^ CFU/mL, for the samples using the NaCl and NaNO_3_ salts, respectively, whereas a significant (*p* < 0.001) reduction in bacterial adhesion was obtained on EO-doped scaffolds (pored with NaCl), particularly 4.24 × 10^4^ ± 7.63 × 10^3^ CFU/mL for the 30% eugenol-added PCL and 3.57 × 10^4^ ± 2.64 × 10^3^ CFU/mL for the 30% cinnamon oil-added PCL. A similar significant (*p* < 0.001) reduction adhesion pattern was obtained for both eugenol and cinnamon oil using NaNO_3_ as porous agent reaching values of 4.13 × 10^4^ ± 1.30 × 10^3^ CFU/mL and 2.57 × 10^4^ ± 8.61 × 10^3^ CFU/mL for the 30% eugenol- and for the 30% cinnamon oil-added PCL, respectively.

Considering *S. epidermidis*—after 7 h of incubation—on the pure-PCL constructs, the data revealed an adhesion rate of about 10^5^ CFU/mL (3.10 × 10^5^ ± 1.13 × 10^4^ CFU/mL using NaCl and 2.36 × 10^5^ ± 3.16 × 10^4^ CFU/mL using NaNO_3_) that significantly (*p* < 0.001) decreased to 10^3^ CFU/mL (*p* < 0.001) when the eugenol was added (NaCl: 2.98 × 10^3^ ± 1.55 × 10^2^ CFU/mL; NaNO_3_: 2.96 × 10^3^ ± 4.00 × 10^2^ CFU/mL), and to 10^2^ CFU/mL (*p* < 0.001) when the cinnamon EO was present (NaCl: 4.55 × 10^2^ ± 1.80 × 10^1^ CFU/mL; NaNO_3_: 2.40 × 10^2^ ± 5.00 × 10^1^ CFU/mL) into the PCL-based biomaterials. 

*E. coli*, after 7 h of incubation, revealed an adhesion pattern on the controls of pure-PCL of 3.53 × 10^7^ ± 2.36 × 10^6^ CFU/mL for NaCl and 8.00 × 10^7^ ± 2.10 × 10^6^ CFU/mL for NaNO_3_. In the meantime, a significant (*p* < 0.001) reduction at 10^6^ CFU/mL was obtained for both 30% eugenol or cinnamon oil-added PCL constructs achieving values of 4.16 × 10^6^ ± 2.33 × 10^5^ CFU/mL and of 2.25 × 10^6^ ± 5.00 × 10^5^ CFU/mL for eugenol, and values of 1.57 × 10^6^ ± 5.65 × 10^5^ CFU/mL and of 1.25 × 10^6^ ± 2.50 × 10^5^ CFU/mL for cinnamon oil, with NaCl or NaNO_3_ as porous agent, respectively, indicating a minor effect of both eugenol and cinnamon oil on the Gram-negative bacterium.

The quantification of planktonic bacteria (i.e., those that did not adhere and remained alive in the broth after 7 h of incubation with the scaffold) was conducted to highlight potential EO release from the biomaterials. As stated, only a small eugenol/cinnamon oil release from the scaffolds was demonstrated; this reflects the values obtained by the quantification of planktonic bacteria. Their CFU/mL where 1-log reduced (*p* < 0.05) for all the three bacteria for the EO-added scaffolds, with respect to pure-PCL—in particular, from 10^8^ to 10^7^ CFU/mL for *S. aureus* and *E. coli,* and from 10^7^ to 10^6^ CFU/mL for *S. epidermidis,* except when *S. epidermidis* was in contact with 30% cinnamon oil added-PCL samples where a 2-log decrease (from 10^7^ to 10^5^ CFU/mL) was observed.

In Figure 10, the adhesion trend—after 24 h of incubation—of *S. aureus* (Figure 10A), *S. epidermidis* (Figure 10B) and *E. coli* (Figure 10C) is reported. Regarding *S. aureus*, a significant (*p* < 0.001) reduction into adhered bacteria was highlighted from control materials with respect to the EO-added PCL constructs (using NaCl as pore former salt): 1.73 × 10^9^ ± 2.58 × 10^8^ CFU/mL for pure-PCL, 2.79 × 10^7^ ± 9.31 × 10^6^ CFU/mL for the 30% eugenol-added PCL and 2.86 × 10^7^ ± 1.01 × 10^6^ CFU/mL for the 30% cinnamon-added PCL. A similar significant (*p* < 0.001) reduction adhesion was obtained for both eugenol and cinnamon oil using NaNO_3_ as a porous agent (Figure 10A). In addition, *S. epidermidis*, after 24 h of incubation, (Figure 10B) revealed a 2-log adhesion reduction (*p* < 0.001) for eugenol-enriched PCL specimens, and a 3-log one (*p* < 0.001) for the cinnamon oil-added PCL scaffolds, with respect to pure PCL, as control, and independently from the salt used to form the pore into the 3D constructs. Regarding *E. coli*, a significant (*p* < 0.001) decrease of the adhesion values was obtained when either eugenol or cinnamon oil at 30% were added into the PCL-based scaffolds, with respect to control materials (from 10^9^ CFU/mL to 10^8^ CFU/mL), indicating a small effect of eugenol and cinnamon oil on the Gram-negative adhesive capability (Figure 10C). 

After 24 h of incubation, the quantification of planktonic bacteria was conducted, and the results are presented in Figure 11: a similar significant (*p* < 0.05 or *p* < 0.001) reduction in planktonic bacteria was recorded, with a count decrease depending on the tested strain.

It is important to highlight that the 30% EO-added samples (independently from the pore-former salt used), with respect to pure PCL-constructs, were able to inhibit not only the adhesion but also the biofilm formation of the three different microorganisms on the scaffolds, as reported in Figure 12 and Figure 13. It is worth noting that the FESEM analysis highlighted that the few bacteria present in the 30% EO enriched PCL-based scaffolds had a relevant morphological alteration, displaying a shape very different from that of the spherical one for staphylococci (Figure 14A), and an enlargement in the bacillary form for *E. coli* (Figure 14B), probably related to a direct effect of the EO on the bacterial cell. 

## 4. Discussion

Microbial resistance to antibiotics has become a relevant issue in the cure of infections and results in the exploration of innovative antibacterial strategies for tissue regeneration applications as well [37]. In this regard, EOs have been pointed out as a suitable approach due to both bactericidal properties and failure to determine antimicrobial resistance [2,10]. Numerous EOs are now available to be used in medicine, but eugenol and cinnamon oil are of major importance [10]. The clove (*Syzygium aromaticum*), belonging to the family *Myrtaceae*, is originated from eastern Indonesia, the EO is extracted from the flower buds, and its major compound is eugenol, a volatile phenylpropanoid, widely used in the pharmaceutical industry [9,29,38]. Cinnamon (*Cinnamomum* spp., *Lauraceae* family) includes a huge number of evergreen trees present in Asia, China, and Australia. Cinnamon EO derived from the different parts of the tree, such as the leaves, bark, fruits, root bark, flowers, and buds [39]. The main components of cinnamon EO are cinnamaldehyde, eugenol, phenol, and linalool [38,40]. Both eugenol and cinnamon oil are known for their properties, such as anti-inflammatory and anti-microbial [9,38,40]. Unfortunately, the concentration and the particular mixture of EOs are essential for their cytotoxicity and consequently for their application in the medical field: EOs at high concentrations may be cytotoxic for both eukaryotic and microbial cells [25], thus an appropriate tuning is strictly necessary. 

In orthopedic surgery, specifically when a prosthetic device is infected due to primary or revision procedures, prosthetic joint infections (PJIs) occur, and, nowadays, the Gram-positive bacteria are the most recovered pathogens, with staphylococci accounting for ~60–80% of infections. Primarily, *S. epidermidis* and *S. aureus* are isolated, even if other Gram-positive bacteria (i.e., *Enterococcus* spp. and *Streptococcus* spp.) could be also found [18,19,41,42,43]. Notably, Gram-negative or anaerobic bacteria and fungi can also determine a PJI despite being rare [18,44,45].

The choice of PCL as synthetic polymer was due to the outstanding properties of this biomaterial, as reported by literature data. PCL, as bioresorbable hydrophobic material, is now recommended for tissue engineering, mainly in orthopedic regenerative medicine, as filler for bone loss as a porous scaffold that allows tissue reconstruction, permitting the targeted delivery of molecules, that are antimicrobials, metal ions, trophic factors, and other compounds [30,37,46,47,48,49,50]. Several studies on PCL and EOs have been performed and are actually available; however, they mainly focused on fibers for small tracts of bone/cartilage or vascular reconstruction to infection prevention, on fibers for wound healing, or on films for food packaging and foodborne disease control [1,4,22,51,52,53]. However, research data about the activity of EOs in combination with PCL as 3D constructs, evaluating their properties in orthopedic regenerative medicine, are still lacking. This knowledge is particularly important for the designing of a more tailored 3D scaffold able to support tissue regeneration and, simultaneously, regret microbial growth and infection within the device implantation site [43]. Our previous work revealed that PCL-based biomaterials enriched with 1.67% of silver were able to counteract *S. aureus* adhesion and biofilm production, but they did not reach non-toxic effects on eukaryotic cells [15,30]. 

To the best of our knowledge, in the present research, for the first time, a PCL-based 3D scaffold was successfully loaded with two different EOs, eugenol and cinnamon oil, at increasing concentrations (from 30 to 50%) to achieve anti-adhesive and anti-biofilm properties against three pathogens involved in biomaterial associated infections, which are *S. aureus*, *S. epidermidis* and *E. coli*. Outstandingly, the PCL-based biomaterials were not toxic for eukaryotic Saos-2 cells only when the lowest EO concentration (30% eugenol or 30% cinnamon oil) was added to the polymer.

From the morphological point of view, by FESEM analysis, the PCL-based constructs were characterized by two different types of pores highly interconnected: the macropores created by the dissolution of salt granules, and the additional (micro)pores ascribed to acetone evaporation. It is to be underlined that the addition of eugenol and cinnamon oil to the PCL polymer did not affect the overall architecture of the 3D scaffolds. The presence of these molecules in the construct only determined an increase into their weight and density, with respect to the pure PCL, used as controls, but only at 40% and 50%. The degradation rate assay was performed by soaking the PCL-based scaffolds, with and without eugenol and cinnamon oil (at increasing percentages), in SBF at 37 °C within 18 days of immersion to evaluate the weight loss during time. The results highlighted that the constructs of pure-PCL and 30%-added PCL based scaffolds reached a small loss in weight over time, whereas a significant (*p* < 0.05 or *p* < 0.001) weight loss was obtained for the 40–50%-enriched samples, probably due to the oil release from the 3D scaffolds. PCL, as bioresorbable synthetic polymer, is a good 3D platform for natural bone-tissue reconstruction—to permit the regeneration of defective and damaged areas—in which a long-term degradation kinetic is required, thus the low degradation rate obtained here further confirm this important specificity of PCL [35,54]. 

An additional relevant challenge in regenerative medicine is the choice of an antimicrobial agent able to counteract microbial adhesion and biofilm formation into a biocompatible material. In this context, we successfully added different and increasing (from 30 to 50%) concentrations of eugenol and cinnamon oil on the PCL-based scaffolds. No cytotoxic effect was observed here with pure PCL, as also previously determined [30], and with the PCL scaffolds added with the lowest concentration of EOs (30%). In fact, at six days of incubation, ~75% and 78% of viability were recorded for the 30%-enriched PCL and pure PCL constructs, respectively, with respect to <40% of the 40–50%-added PCL samples. After 12 days of incubation control material and 30% eugenol and cinnamon oil-loaded samples maintained a viability of ~70%, whereas a further significant decrease into viability was demonstrated for 40 and 50%-enriched scaffolds achieving values <10%. To guarantee non-lethal treatment, a cell viability of 70% or greater was considered adequate as formerly established (“ISO 10993–5:2009-Biological evaluation of medical devices–Part 5: Tests for in vitro cytotoxicity”). The non-toxic behavior of pure PCL demonstrated here is in good agreement with Yun et al. [54]. The FESEM analysis, conducted on the 30% eugenol or cinnamon oil enriched samples in the presence of Saos-2 cells, revealed that, after six days of incubation, the eukaryotic cells attach the biomaterial; thereafter, at 12 days of incubation, osteoblast-like cells further integrated into the 3D scaffolds. 

In a study of Junka and colleagues [22], hydroxyapatite coated with microfibrils of bacterial cellulose impregnated with EOs was prepared—with the aim to reduce bone infection—and assayed for cytotoxicity against both osteoblasts and fibroblasts. They revealed that the highest survival rate was obtained for the eucalyptus EO, with respect to clove and thyme EOs, for both types of cells. In another work, the authors fabricated bioactive glass/soy protein composite scaffolds covered with cinnamon EO and observed that osteoblast-like cell viability was reduced at the higher EO concentrations, whereas a non cytotoxic effect was detected at the lower concentrations of the cinnamon oil [42]. In addition, in another research study, an EO concentration effect on eukaryotic cell viability was proved, as we also did [9].

For these reasons, the microbiological experiments were performed only assaying the non-toxic 30% concentration of eugenol and cinnamon oil. The process to load the PCL with the eugenol and cinnamon oil was able to have these compounds mixed and integrated with the polymer. This property of the PCL-based scaffolds added with the 30% eugenol and cinnamon oil EOs was evident considering the microbiological results. Regarding the inhibition halo tests, we observed that the inhibition of the bacterial growth around the 30% eugenol or 30% cinnamon oil PCL-based constructs was present only for the Gram-positive bacteria (both *S. aureus* and *S. epidermidis*) reaching higher diameter values for cinnamon oil with respect to eugenol, whereas no inhibition halo was obtained for the Gram-negative one, *E. coli*. The adhesion experiments demonstrated that the initial adhesion pattern to a biomaterial is strongly influenced not only by the bacterial genus, but also by the species, so it is strictly dependent on the microorganisms, as we demonstrated in previous research too [55,56,57]. After 7 h of incubation, the adhesion rate to pure-PCL was on the order of 10^5^ CFU/mL for *S. epidermidis*, of 10^6^ CFU/mL for *S. aureus*, and of 10^7^ CFU/mL for *E. coli*, which increased to 10^6^, 10^7^ and 10^8^ CFU/mL, respectively, at 24 h of incubation. Notably, the addition of 30% eugenol and 30% cinnamon oil to the PCL constructs determined a significant (*p* < 0.001) 2-log reduction into adhered bacteria independently from the EO (eugenol or cinnamon oil) or the porous agent (NaCl or NaNO_3_), even if a 3-log reduction was observed for *S. epidermidis* adhesion into the 30% cinnamon added 3D scaffolds. In the meantime, the count of the planktonic bacteria growth in contact with the 30% EO-added PCL specimens revealed a 1-log decrease into the CFU/mL results, with values varying according to the bacterial strain assayed. Only for *S. epidermidis* was a 2-log reduction obtained for the bacteria at direct contact to the 30% cinnamon oil enriched PCL specimens. These results seem to confirm that only a small portion of eugenol and cinnamon oil was released by the constructs, maybe the one not fully bonded into the polymer matrix due to the complex nature of these molecules. In addition, since the cinnamon EO used in the present work is a complex mixture of compounds—as also referred by several authors [10,39]—whereas eugenol is a >98% pure molecule, this might explain its more pronounced effect, mainly against *S. epidermidis*.

Teles et al. [9] demonstrated that eugenol is more effective against *S. aureus* with respect to *E. coli*, by inhibition halo assays, as we also did, but, in our research, the EO was included into the polymer scaffold, whereas other authors revealed a direct good anti-*E.coli* activity of clove—from which eugenol derived—with both zone inhibition and minimum inhibitory concentrations (MICs), also against resistant strains [8] and against various microorganisms [10,38,40,58,59], thus confirming that the EO composition is an important issue for its anti-bacterial activity. Regarding cinnamon oil, some authors confirmed both anti-bacterial and anti-biofilm actions against Gram-negative and Gram-positive bacteria, and, once again, with a more pronounced action on the latter [10,38,39,40,42,58,60]. These results are consistent with our data. This could be explained by the different composition and architecture of the two types of bacteria, where the absence of an outer cell lipid layer and periplasm allows higher molecule penetrability in the Gram-positive bacteria, while the Gram-negative are capable of restricting diffusion of hydrophobic molecules through its lipopolysaccharide envelope reducing penetration [4,10,12,29,38,39,60]. The FESEM analysis, conducted in the present work, further confirmed a direct action of the EOs added to the PCL-based constructs as a relevant morphological change occurred in bacteria, in line with data from other authors [2,10,29,39,42]. 

We also indirectly demonstrated an anti-biofilm/anti-aggregates action of the PCL based 3D scaffolds containing EOs. In fact, by FESEM observation, we highlighted that only the PCL-based biomaterials enriched with eugenol or cinnamon oil were able to counteract biofilm (for both *S. aureus* and *S. epidermidis*) or bacterial cell aggregates (for *E. coli*) formation with respect to pure-PCL, used as control material. Our findings are in agreement with the results of other research [29,38,39,40].

It is to be highlighted that the GC-MS analysis revealed that the active concentration of the EOs within the 3D constructs was reduced, achieving ~10% of eugenol or cinnamon oil content, in the 30% EO-doped sample. These data agree well with other literature [22,42], where concentrations <10% of EO were incorporated into a biomaterial that demonstrated both non-toxic behavior to eukaryotic cells, and, in parallel, anti-bacterial and anti-biofilm actions. 

The microbiological results altogether support the knowledge that, if an antimicrobial agent, such as an EO or a single component, is successfully loaded into a biomaterial, specifically the polymer PCL as 3D construct, the initial microbial adhesion is prevented, and the succeeding phase of biofilm or bacterial cell aggregate development would also be inhibited. Moreover, the addition to PCL of the lowest concentration of eugenol or cinnamon oil is additionally able to promote osteoblast cell proliferation and viability. 

## 5. Conclusions

Despite biomedical devices having improved regenerative medicine, they can also be associated with the adverse effect of the raised risk of infections by pathogens able to resist conventional antimicrobial agents, determining to be of primary importance the research of new antibacterial strategies. Recently, EO based approaches emerged as a promising candidate thanks to their antimicrobial activities and incapability to induce microbial resistance.

In summary, we reported for the first time the design and development of an effective 3D platform based on a polycaprolactone polymer containing eugenol and cinnamon oil, with a highly porous structure, non-toxic behavior, and anti-adhesive properties against Gram-positive and Gram-negative bacteria. The biodegradability and biocompatibility of the PCL together with bioactive natural compounds (EOs) make these 3D scaffolds as eco-friendly anti-adhesive/anti-biofilm constructs, granting the targeted delivery of the EOs into the site of infection where bacteria are allocated, and might have great potential applications in the regenerative medicine field. The present research offers the sound basis for further studies that aimed to improve the biomaterials with calcium phosphates to confer to them bioactive properties and to explore the antifungal effects of the EO-enriched PCL based 3D scaffolds too.

## Figures and Tables

**Figure 1 pharmaceutics-14-01873-f001:**
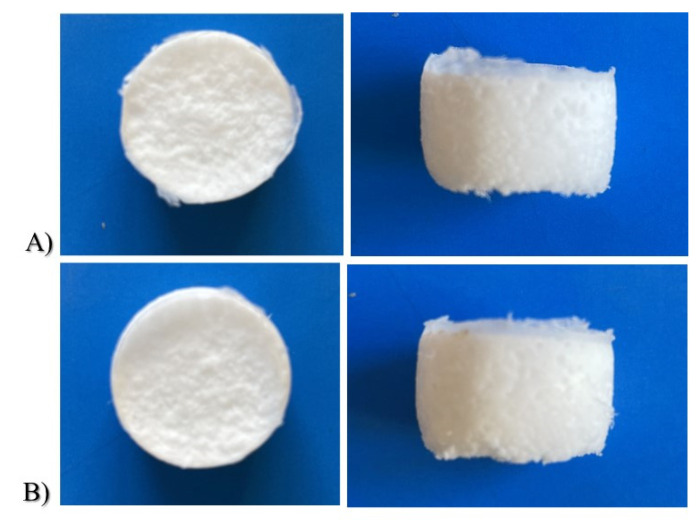
Representative images of the general aspect of the PCL-based scaffolds: pure PCL (**A**) and 30% EO-added PCL (**B**) samples with NaCl.

**Figure 2 pharmaceutics-14-01873-f002:**
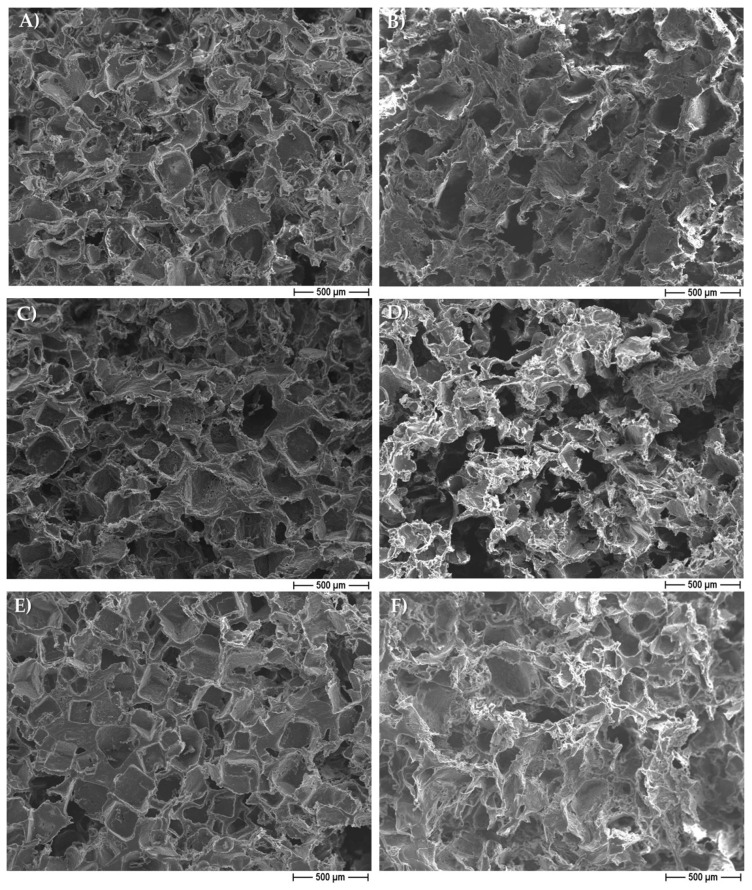
Field emission scanning electron microscopy (FESEM) micrographs of pure poly(ε-caprolactone) (PCL) scaffolds obtained by using sodium chloride, NaCl (**A**), and sodium nitrate, NaNO_3_ (**B**) salts as templates; micrographs of 30% eugenol-added (**C**,**D**) and 30% cinnamon oil-added (**E**,**F**) PCL constructs, showing the same morphology of the neat polymer scaffold.

**Figure 3 pharmaceutics-14-01873-f003:**
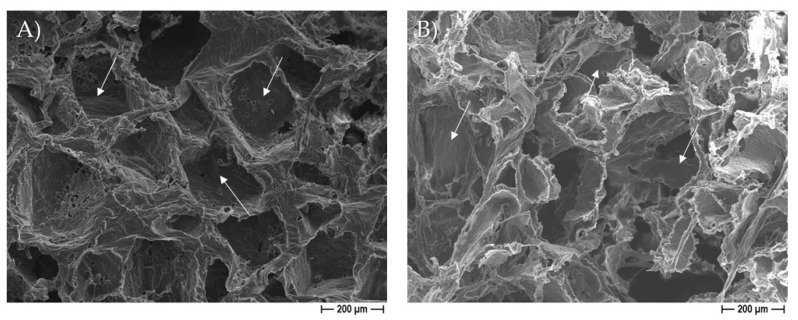
Representative higher magnification FESEM micrographs of PCL samples obtained by adding 30% eugenol EO to the polymer using NaCl (**A**) or 30% cinnamon oil to the polymer using NaNO_3_ (**B**) as pore forming salt, showing microporosities within the struts, and on the pore walls (white arrows).

**Figure 4 pharmaceutics-14-01873-f004:**
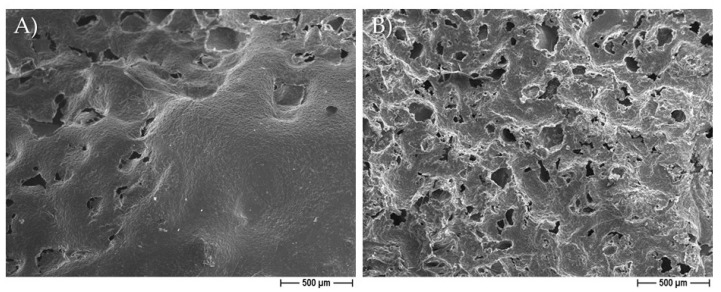
Representative FESEM micrographs of PCL samples obtained by adding 30% eugenol to the polymer using NaCl (**A**) or NaNO_3_ (**B**) as pore forming salt, showing the surface microporosities.

**Figure 5 pharmaceutics-14-01873-f005:**
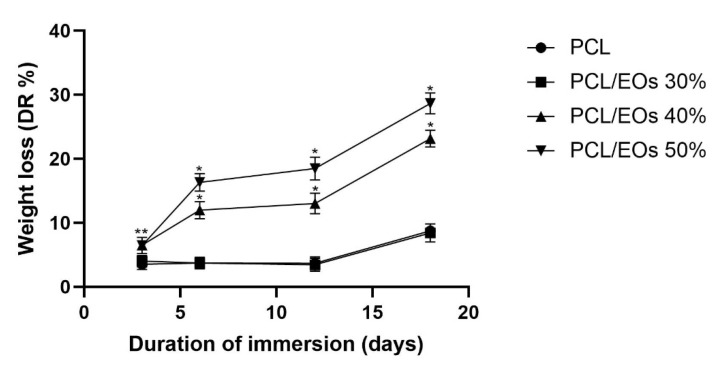
Weight loss, expressed as degradation rate % (DR%), of the pure PCL and EOs-added PCL-based 3D scaffolds at different time points (3, 6, 12 and 18 days) after the immersion in SBF. ** *p* < 0.05 or * *p* < 0.001 vs. PCL and 30%-added PCL based scaffolds, unpaired *t*-test.

**Figure 6 pharmaceutics-14-01873-f006:**
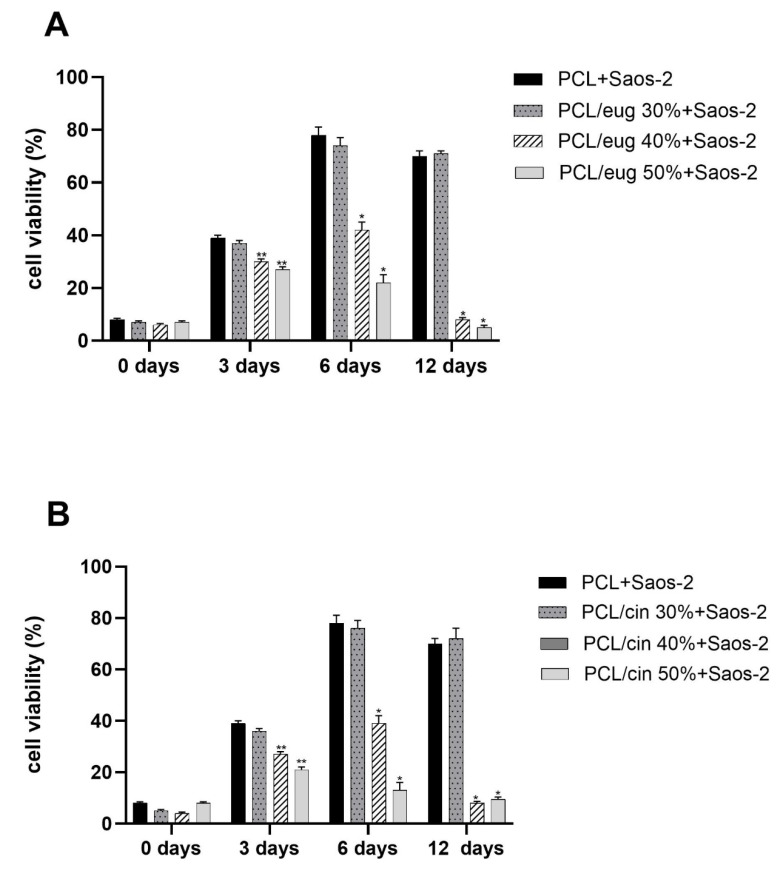
Cell viability (3-(4,5-Dimethylthiazol-2-yl)-2,5-Diphenyltetrazolium Bromide, MTT analysis) of sarcoma osteogenic-2 (Saos-2) cells exposed to PCL-based biomaterial enriched with eugenol (**A**) or cinnamon oil (**B**) at increasing concentrations (from 30% to 50%), expressed as cell viability percentages (%). Results are the mean values ± standard error of the mean (SEM) of at least three independent experiments; ** *p* < 0.05 or * *p* < 0.001 vs. PCL + Saos-2, PCL/eug 30% + Saos-2 and PCL/cin 30% + Saos-2, unpaired *t*-test.

**Figure 7 pharmaceutics-14-01873-f007:**
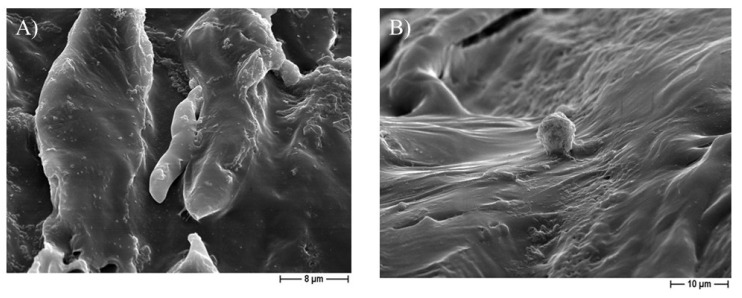
Representative FESEM micrographs, after six days of incubation, of the 30% cinnamon oil-added PCL 3D scaffolds obtained by using NaCl (**A**) or NaNO_3_ (**B**) salt as template, showing the attachment of Saos-2 cells at 1000× or 2000× magnification, respectively.

**Figure 8 pharmaceutics-14-01873-f008:**
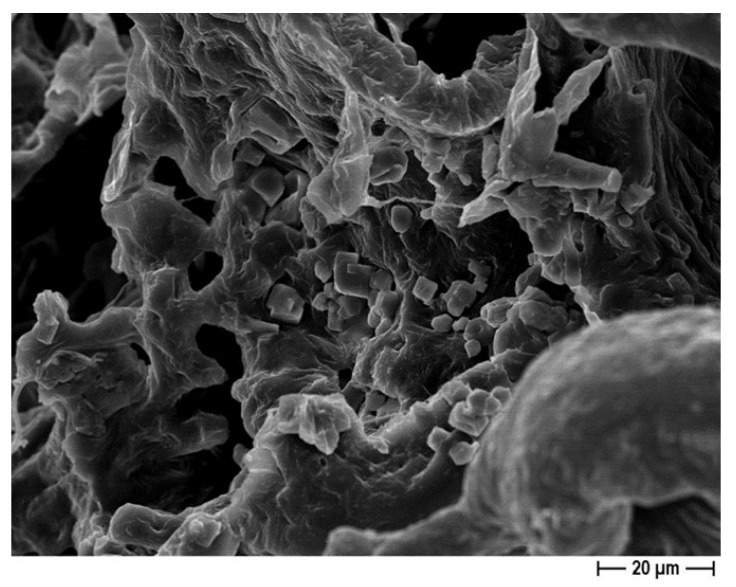
Representative FESEM micrographs of the 30% cinnamon oil-added PCL 3D scaffolds obtained by using NaCl salt as template, showing the integration of different Saos-2 cells at 1000× magnification, after 12 days of incubation.

**Figure 9 pharmaceutics-14-01873-f009:**
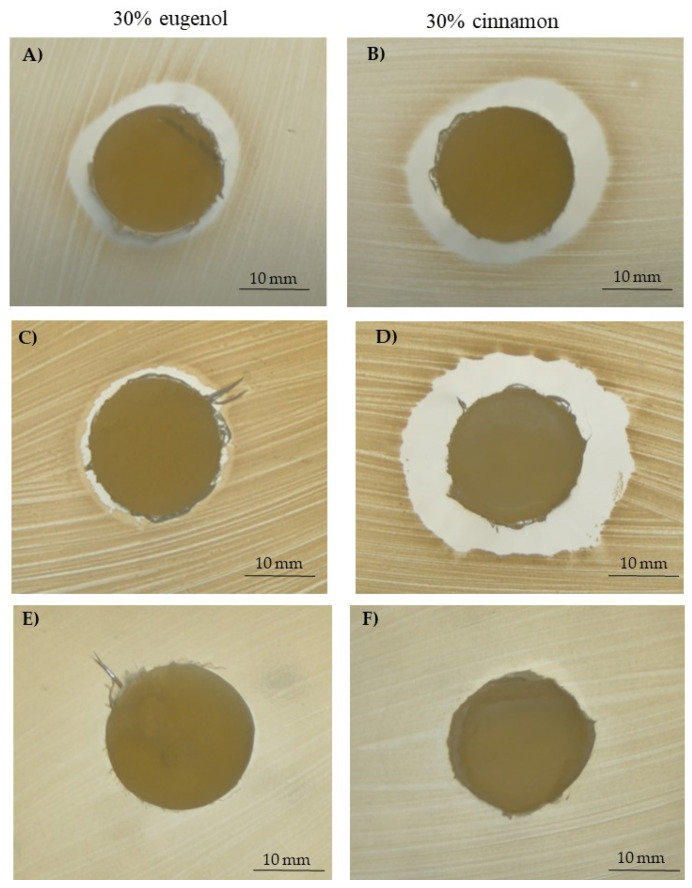
Representative images of the inhibition halo test: PCL-based samples enriched with 30% eugenol or 30% cinnamon oil towards *S. aureus* (**A**,**B**), *S. epidermidis* (**C**,**D**) and *E. coli* (**E**,**F**).

**Figure 10 pharmaceutics-14-01873-f010:**
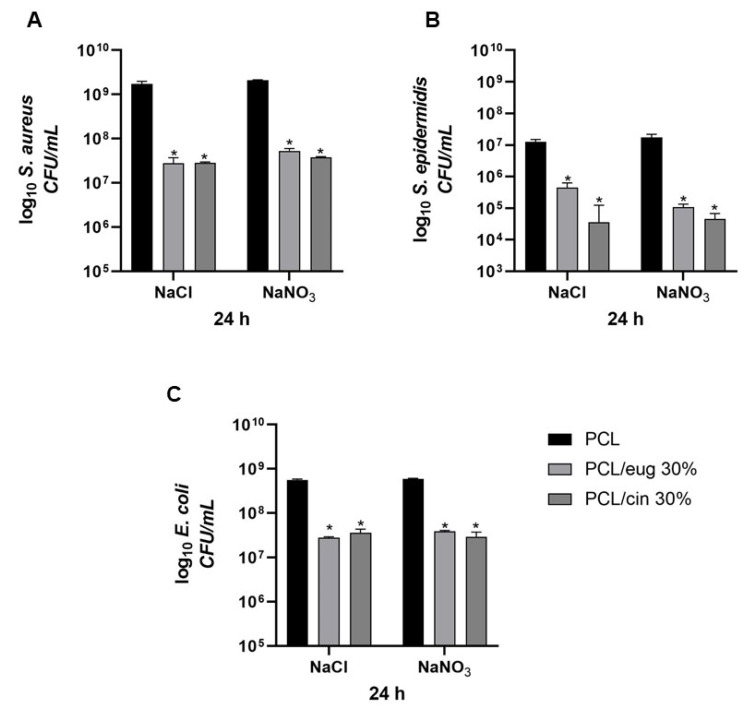
Number of adherent *S. aureus* (**A**), *S. epidermidis* (**B**) and *E. coli* (**C**) (log_10_ colony forming unit, CFU/mL) on the PCL scaffolds added with 30% eugenol or cinnamon oil, produced with either NaCl or NaNO_3_ salts, after 24 h of incubation. Results are the mean values ± standard error of the mean (SEM) of at least three independent experiments. * *p* < 0.001 vs. PCL, unpaired *t*-test.

**Figure 11 pharmaceutics-14-01873-f011:**
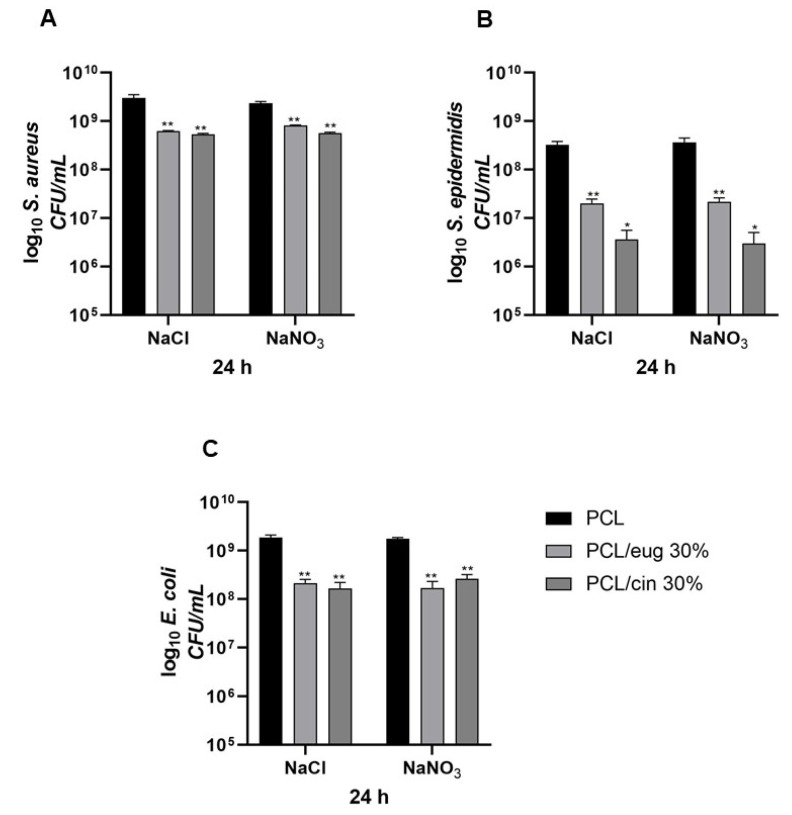
Number of planktonic *S. aureus* (**A**), *S. epidermidis* (**B**) and *E. coli* (**C**) (log_10_ colony forming unit, CFU/mL) on the PCL scaffolds added with 30% eugenol or cinnamon oil, produced with either NaCl or NaNO_3_ salts, after 24 h of incubation. Results are the mean values ± standard error of the mean (SEM) of at least three independent experiments. ** *p* < 0.05 or * *p* < 0.001 vs. PCL, unpaired *t*-test.

**Figure 12 pharmaceutics-14-01873-f012:**
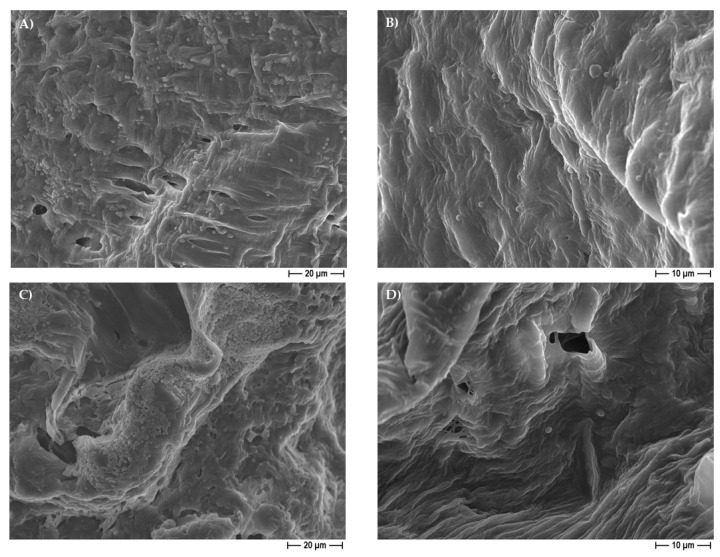
Representative FESEM micrographs regarding *S. aureus* presence on non-sonicated PCL-based scaffolds (**A**) and 30% cinnamon oil-added PCL-based scaffolds (**B**), or *S. epidermidis* presence on non-sonicated PCL-based scaffolds (**C**) and 30% cinnamon oil-added PCL-based scaffolds (**D**), obtained by using NaCl salt as a template, at 1000× or 2000× magnification, respectively.

**Figure 13 pharmaceutics-14-01873-f013:**
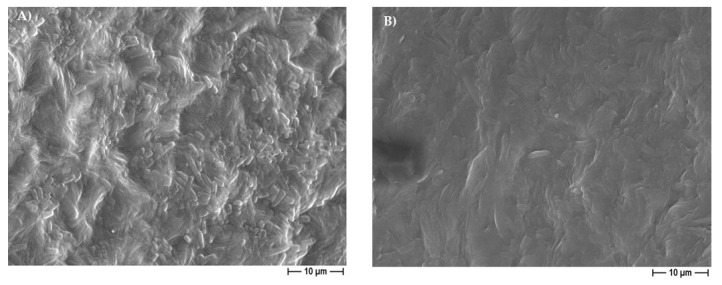
Representative FESEM micrographs regarding *E. coli* presence on non-sonicated PCL-based scaffolds (**A**) and 30% eugenol-added PCL-based scaffolds (**B**), obtained by using NaCl salt as a template, at 2000× magnification.

**Figure 14 pharmaceutics-14-01873-f014:**
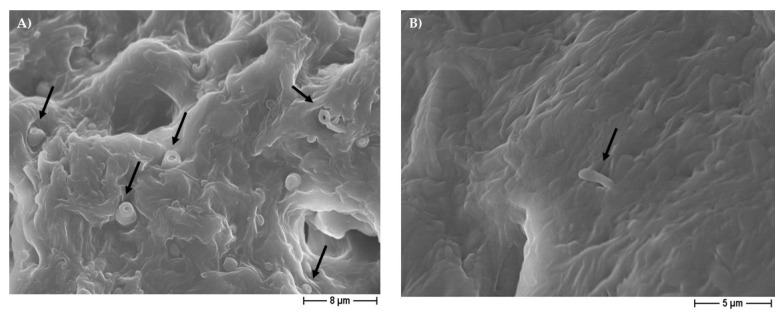
Representative FESEM micrographs of *S. aureus* (**A**) morphological alteration (black arrows) on 30% cinnamon oil PCL-based scaffolds (3000× magnification), and of *E.coli* (**B**) morphological enlargement (black arrow) on 30% eugenol PCL-based scaffolds, obtained by using NaCl salt as a template (5000× magnification).

**Table 1 pharmaceutics-14-01873-t001:** Morphological characteristics (reported as mean ± standard error of the mean) of the PCL-based scaffolds (obtained with NaCl) added with increasing (from 30 to 50%) concentrations of eugenol (**A**) or cinnamon oil (**B**).

A	Morphological Parameters				Statistical Analysis
	Diameter (mm)	Height (mm)	Weight (mg)	Density (mg/mm^3^)	Student’s *t*-Test
Scaffold Type					
PCL	18.31 ± 0.11	11.36 ± 0.14	377.6 ± 10.38	0.126 ± 0.003	weight and density PCL vs. PCL + eug 40% and 50% *p* < 0.001
PCL + eug 30%	17.64 ± 0.12	11.91 ± 0.34	421.8 ± 14.60	0.133 ± 0.005	
PCL + eug 40%	17.88 ± 0.06	10.66 ± 0.12	809.5 ± 13.35	0.303 ± 0.011	
PCL + eug 50%	17.40 ± 0.20	9.29 ± 0.27	689.2 ± 13.00	0.301 ± 0.008	
**B**					
PCL	18.31 ± 0.11	11.36 ± 0.14	377.6 ± 10.38	0.126 ± 0.003	weight and density PCL vs. PCL + cin 40% and 50% *p* < 0.001
PCL + cin 30%	17.95 ± 0.09	11.69 ± 0.45	377.8 ± 8.52	0.129 ± 0.005	
PCL + cin 40%	18.09 ± 0.04	9.88 ± 0.23	801.7 ± 2.35	0.310 ± 0.010	
PCL + cin 50%	17.25 ± 0.08	9.29 ± 0.27	600.4 ± 13.15	0.285 ± 0.100	

Abbreviations. PCL: poly(ε-caprolactone); NaCl: sodium chloride; Eug: eugenol; Cin: cinnamon oil.

**Table 2 pharmaceutics-14-01873-t002:** Morphological characteristics (reported as mean ± standard error of the mean) of the PCL-based scaffolds (obtained with NaNO_3_) added with increasing (from 30 to 50%) concentrations of eugenol (**A**) or cinnamon oil (**B**).

A	Morphological Parameters				Statistical Analysis
	Diameter (mm)	Height (mm)	Weight (mg)	Density (mg/mm^3^)	Student’s *t*-Test
Scaffold Type					
PCL	18.10 ± 0.13	10.17 ± 0.15	366.7 ± 9.55	0.136 ± 0.002	weight and density PCL vs. PCL + eug 40% and 50% *p* < 0.001
PCL + eug 30%	17.74 ± 0.12	12.00 ± 0.56	425.0 ± 7.01	0.163 ± 0.007	
PCL + eug 40%	17.60 ± 0.09	11.52 ± 0.19	822.6 ± 10.22	0.319 ± 0.015	
PCL + eug 50%	17.81 ± 0.14	9.91 ± 0.28	653.8 ± 11.10	0.297 ± 0.003	
**B**					
PCL	18.10 ± 0.13	10.17 ± 0.15	366.7 ± 9.55	0.136 ± 0.002	weight and density PCL vs. PCL + cin 40% and 50% *p* < 0.001
PCL + cin 30%	17.46 ± 0.07	11.4 1 ± 0.51	428.8 ± 5.81	0.161 ± 0.007	
PCL + cin 40%	18.33 ± 0.08	10.22 ± 0.17	831.1 ± 8.34	0.308 ± 0.011	
PCL + cin 50%	17.95 ± 0.10	9.89 ± 0.26	674.1 ± 9.66	0.305 ± 0.013	

Abbreviations. PCL: poly(ε-caprolactone); NaNO_3_: sodium nitrate; Eug: eugenol; Cin: cinnamon oil.

## Data Availability

The source data underlying tables and figures are available from the authors upon request.

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
