# Peer review of "Combination of Poly(ε-Caprolactone) Biomaterials and Essential Oils to Achieve Anti-Bacterial and Osteo-Proliferative Properties for 3D-Scaffolds in Regenerative Medicine"

_pharmaceutics, 2022, doi:10.3390/pharmaceutics14091873_

Round 1

Reviewer 1 Report

Comments:

In this manuscript, Sara Comini and colleagues investigated the composition and composition of essential oils and combined PLC and eugenol with the antibacterial properties of cinnamon oil, but the data on the content of this work were not complete.

1.The whole manuscript picture is too redundant and several of them can be appropriately condensed into one 

2.In antibacterial experiments, the size of the inhibition zone was not statistically analyzed graphically, and the data were not intuitively displayed.

3.Why NaCI and NaNO3 were chosen in bacterial colony assays, is it arbitrary? such as CaSO4?

4.In experiments of cell survival, PCL/cin 30% + Saos-2 had much higher cell survival than PCL/cin 40% + Saos-2 from day 6 onwards, and whether it was possible to eliminate samples at 50% concentration to add a new gradient at 30% – 40% ?

5. Cell survival was compared between eugenol and cinnamon oil at different concentrations in the experiments above, why only 30% of the groups were listed in the bacterial colony experiments?

Reviewer 2 Report

Overall, this is a clear and well-written manuscript. The main objective of the present research was to design and develop novel modified 90 PCL 3-dimensional (3D) scaffolds - for potential applications in the biomedical field - with 91 antimicrobial properties by adding eugenol and cinnamon EO, as antimicrobial agents.This study has some clinical significance. Nonetheless, the manuscript can be further improved and the following concerns should be adequately addressed. My detailed comments are as follows:

1. Please add the general view of the scaffolds.

2. Please compare the porosity of the scaffolds.

3. These are many similar studies, please compare the differences.

4. In the figure 5, please add the resutls of osteogenic genes expression

Round 2

Reviewer 2 Report

Congratulation!